# Anaerobic Storage Completely Removes Suspected Fungal Pathogens but Increases Antibiotic Resistance Gene Levels in Swine Wastewater High in Sulfonamides

**DOI:** 10.3390/ijerph20043135

**Published:** 2023-02-10

**Authors:** Xinyue Zhao, Mengjie Zhang, Zhilin Sun, Huabao Zheng, Qifa Zhou

**Affiliations:** 1College of Life Sciences, Zhejiang University, Hangzhou 310058, China; 2College of Architecture Engineering, Zhejiang University, Hangzhou 310058, China; 3Zhejiang Province Key Laboratory of Soil Contamination Bioremediation, Zhejiang A&F University, Hangzhou 311300, China

**Keywords:** antibiotic resistance genes, pathogens, sulfonamides, swine wastewater, wastewater storage

## Abstract

Wastewater storage before reuse is regulated in some countries. Investigations of pathogens and antibiotic resistance genes (ARGs) during wastewater storage are necessary for lowering the risks for wastewater reuse but are still mostly lacking. This study aimed to investigate pathogens, including harmful plant pathogens, and ARGs during 180 d of swine wastewater (SWW) storage in an anaerobic storage experiment. The contents of total organic carbon and total nitrogen in SWW were found to consistently decrease with the extension of storage time. Bacterial abundance and fungal abundance significantly decreased with storage time, which may be mainly attributed to nutrient loss during storage and the long period of exposure to a high level (4653.2 μg/L) of sulfonamides in the SWW, which have an inhibitory effect. It was found that suspected bacterial pathogens (e.g., *Escherichia–Shigella* spp., *Vibrio* spp., *Arcobacter* spp., *Clostridium_sensu_stricto_1* spp., and *Pseudomonas* spp.) and sulfonamide-resistant genes *Sul1*, *Sul2*, *Sul3*, and *SulA* tended to persist and even become enriched during SWW storage. Interestingly, some suspected plant fungal species (e.g., *Fusarium* spp., *Ustilago* spp. and *Blumeria* spp.) were detected in SWW. Fungi in the SWW, including threatening fungal pathogens, were completely removed after 60 d of anaerobic storage, indicating that storage could lower the risk of using SWW in crop production. The results clearly indicate that storage time is crucial for SWW properties, and long periods of anaerobic storage could lead to substantial nutrient loss and enrichment of bacterial pathogens and ARGs in SWW.

## 1. Introduction

Swine wastewater (SWW) contains high concentrations of ammonia, particulate organic matter (POM), and dissolved organic matter (DOM) in addition to diverse species of microbes [1,2,3,4]. Of the microbes found in SWW, the possible bacterial pathogens (e.g., *Escherichia* spp., *Arcobacter* spp., *Clostridium* spp.) [2] and possible fungal pathogens (e.g., *Aspergillus* spp., *Trichosporon* spp. and *Cladosporium* spp.) [5] are of great concern. Additionally, virus pathogens (e.g., Porcine circovirus type 2) have also been detected in SWW [3]. In particular, high concentrations of antibiotics and antibiotic resistance genes (ARGs) are frequently detected in SWW in China [6,7,8,9], as approximately 4.5 million kg sulfonamides and 16 million kg tetracyclines are used per year in swine production in China [10]. SWW effluent can be reused as liquid manure [11] and irrigation water [12], as it contains valuable amounts of water and nutrients. However, reuse of SWW could result in the delivery of pathogens [2] and ARGs [8] into the recipient environment. The pathogens in inadequately treated SWW can contaminate the derived food items, soil, water, and re-infect humans and animals [13,14], representing a potential “One Health” risk, i.e., to humans, animals, and environmental health [14,15]. Furthermore, SWW can contain pathogens harmful to crops, which could seriously reduce crop yield and quality if SWW is reused as liquid manure and irrigation water. Generally, effluents should be stored for a certain period before use, which is regulated in some countries, such as Italy, Finland, Denmark, and China [16]. The required storage times are 60, 180, and 270 days in China, Italy and Finland, and Denmark, respectively [16]. This variation in the required storage time may result from differences in the standard and wastewater properties in these countries. During the storage of ammonia-containing effluent or digestate, complex carbon (C) and nitrogen (N) transformations, including nitrification–denitrification, ammonia oxidation, anaerobic ammonium oxidation and partial nitrification–anaerobic ammonium oxidation, ammonia assimilation, and N fixation [17,18,19,20], could significantly affect the water and fertiliser qualities of the effluent or digestate upon application. Specifically, some N transformations (e.g., volatilisation of ammonia, nitrification–denitrification, and ammonia oxidation) can lead to significant N loss during wastewater storage, which would pollute the environment and decrease the amount of nutrients supplied to plants [19]. Furthermore, storage could significantly alter the microbial community structure in wastewater. Therefore, storage time could crucially influence wastewater properties. However, relevant studies remain rare. Specifically, investigations of effects and influencing mechanisms of storage time on pathogens and ARGs during wastewater storage are particularly needed. In this study, we aimed to investigate the effects and influencing mechanisms of storage time on pathogens and ARGs during SWW storage, which could be helpful for evaluating anaerobic storage as a remediation strategy.

## 2. Materials and Methods

### 2.1. Wastewater

Raw SWW was obtained from a local swine farm in Huzhou City, Zhejiang Province, China. The wastewater was filtrated through 120 μm filtration paper (Shuangquan Co., Hangzhou, China) before use.

### 2.2. SWW Storage Experiment

A batch SWW anaerobic storage experiment was conducted at 30 °C in an enclosed 2.0 mm thick polypropylene plastic container (25 L working volume) from May to November 2020. There were three replicates for the storage treatment. The total storage time was 180 d, and wastewater samples were collected once every 30/60 days during the storage process.

### 2.3. Measurement of Wastewater Properties

Each wastewater sample was filtered through a 0.45 μm cellulose membrane, and the filtrate was collected for analysis. The hydrogen ion concentration (pH) was measured using a PHB-4 pH meter (INESA Co., Shanghai, China). The total organic carbon (TOC) content was measured using a SHIMADZU TOV-Vcph analyser (SHIMADZU Corp., Kyoto, Japan), total N (TN) was determined using the micro-Kjedahl method, and the NH_4_-N content was determined following the Nash reagent spectrophotometric method (SEPA, 2002) [21]. NO_3_-N and NO_2_-N were determined using ion chromatography (Dionex ICS-1500 Ion Chromatography System, SpectraLab Scientific Inc., Markham, ON, Canada) following methods described by SEPA (2002) [21]. SON was calculated by TN subtracting NH_4_-N, NO_3_-N, and NO_2_-N. Sulfonamides were determined using liquid chromatography–mass spectrometry/mass spectrometry (Agilent 1290-6470, Agilent, Palo Alto, CA, USA) (LCMSMS, following AOAC 2007.01) according to the methods of Li et al. (2021) [22].

### 2.4. High-Throughput Sequencing

Swine wastewater (30 mL) was collected from each container on storage days 0, 60, 120, and 180 (SWW_0_, SWW_60_, SWW_120_, and SWW_180_). The TIANNAMP DNA Kit (TIANGEN, Beijing, China) was used to extract DNA. Before DNA extraction, the sample was centrifugated at 300× *g* for 5 min to precipitate large particles. The supernatant was then removed and centrifuged at 19,000× *g* for 5 min to precipitate cells. Moreover, the pellet was washed twice with 3.0 mL of sterile distilled water to reduce inhibitors for *Taq* polymerase. The quantity and quality of DNA were evaluated via 1% gel, NanoDrop 2000 (Thermo Fisher Scientific, Waltham, MA, USA) and Qubit 3.0 (Thermo Fisher Scientific, Waltham, MA, USA), and the DNA was then stored at −80 °C before use. Sample integrity was tested by agarose gel electrophoresis. The 16S rRNA gene of bacteria (in the V3–V4 region) was amplified using the primer pairs 515F and 806R, and 528F and 706R [3]. The rRNA gene of fungi was amplified using the ITS1-F, ITS2-2043R, and ITS5-1737F primers [23]. Sequencing of bacteria and fungi was performed using the Illumina HiSeq 2500 platform (BGI Co., Ltd., Shenzhen, China). Sequence analyses were conducted according to the methods of Wang et al. [3] and Shi et al. (2022) [23].

### 2.5. Quantitative PCR and Quantification of ARGs

Swine wastewater (2 mL) collected from each container on storage days 0, 60, 120, and 180 was extracted using the TIANNAMP DNA Kit (TIANGEN, Beijing, China), according to the abovementioned procedure, and then submitted for quantitative PCR and quantification of ARGs.The wastewater sample volume was in accordance with the standard qPCR procedure at the analysis lab. The 16S rRNA gene and the ITS gene were quantified using a StepOnePlus™ RealTime PCR System (Thermo Fisher, Waltham, MA, USA) and TB Green™ Premix Ex Taq™ II (Tli RNaseH Plus) kit (Takara, Kyoto Japan). The qPCR conditions, primer sequences targeting these genes, and PCR protocol were based on previously reported methods [24]. Each reaction contained 0.8 µL of primers, 1 µL of DNA templates, 5 µL of TB Green Premix Ex Taq II (Takara, Japan), 0.2 µL of ROX Reference Dye (Bio-Rad), and 3 µL of ddH2O (Bio-Rad). Each qPCR reaction was conducted in triplicate. Quantification of ARGs were conducted based on the methods of Pu et al. (2018) [7]. As only sulfonamides were used in the swine farm, only the sulfonamide-resistant genes *Sul1*, *Sul2*, *Sul3*, and *SulA* were assessed in this study.

### 2.6. Data Analyses

Tukey tests were used to compare means with 0.05 set as the significance level. The relative abundance for each taxon was calculated as the operational taxonomic unit (OTU) ratio (%) among the total OTUs.

## 3. Results and Discussion

### 3.1. SWW Properties

As shown in Table 1, the SWW contains high contents of TOC, NH_4_-N, and SAs. The properties are similar to those of SWW from typical large-scale swine farm in China [17].

With the extension of the storage time, the SWW TOC steadily decreased (Figure 1) and the TN and NH_4_-N (Figure 2) consistently and significantly (*p* < 0.05) decreased while the NO_3_-N and NO_2_-N contents (Figure 1) remained very low and barely changed and the soluble organic N (SON) content (Figure 1) significantly increased (*p* < 0.05), indicating substantial nutrient loss during SWW storage. The C and N losses can be attributed to microbial transformations [17,18,19,20].

### 3.2. Microbial Abundance and Level of ARGs during SWW Storage

Both the bacterial (Figure 3a) and fungal (Figure 3b) abundances significantly decreased with the storage time. In particular, the fungal abundance decreased to zero after 60 d of storage, indicating that fungi were completely removed from the SWW. The decrease in bacterial and fungal abundances can be mainly attributed to the loss of nutrients (e.g., C and N) and the long period of exposure to inhibitory sulfonamides in the SWW under anaerobic conditions. In particular, the fungi could be more sensitive to nutrient loss because fungi have higher nutrient requirements than bacteria [23]. Additionally, fungi could be more seriously inhibited than bacteria by the sulfonamides in the SWW under anaerobic conditions.

The levels of *Sul1* and *SulA* significantly increased with storage time (Figure 4a), while the levels of *Sul2* and *Sul3* were not significantly different across the four storage time points (Figure 4b). The results clearly indicate that ARGs persisted or were even enriched during the anaerobic storage of SWW. Pu et al. (2018) [7] reported that anaerobic digestion enriches antibiotic-resistant bacteria and resistance genes for *sulA*, aminoglycoside, and cefaclor (FCA) in biogas residue. Tao et al. (2014) [6] also demonstrated that the abundances of *sul1* and *sul2* increased after anaerobic digestion in four pig farm wastewater treatment systems. However, Zhang et al. (2021) [9] reported that ARGs, including resistance genes for *sulA*, were efficiently removed from SWW by anaerobic digestion. In general, the reason for the enrichment of ARGs in anaerobic digestion products has not been reasonably explained [7]. We rationalise that ARG hosts (e.g., *Escherichia–Shigella* spp.) could have a competitive advantage over antibiotic-sensitive species in wastewater when there is a high level of antibiotics. Further studies should be conducted to better understand the final fate of ARGs in wastewater.

### 3.3. Microbial Community

Data for relative bacterial abundance at phylum level are presented in Figure 5. At phylum level, *Firmicutes*, *Proteobacteria*, and *Bacteroidota* were found to be dominant in the SWW. The bacterial structure in Figure 5 is similar to that reported by Zhai et al. (2018) [5]. The relative abundance of *Bacteroidota* tended to increase with the storage time, while that for *Firmicutes* decreased and *Proteobacteria* abundance remained high across the storage time (Figure 5). The dominant species with percentage of total OTUs exceeding 1% were *Thauera* (29.8%), *Methanosaeta* (6.4%), and *Anaerovorax* (1.7%); *Halomonas* (6.0%); *Anaerovorax* (18.4), *Nitrosomonas* (7.0%), and *Vulcanibacillus* (5.3%); and *Paenibacillus* (19.0%) and *Anaerovorax* (12.4%) on storage days 0, 60, 120, and 180, respectively. The most dominant species in the raw SWW, *Thauera* sp., is a heterotrophic denitrification species [24], and *Thauera* sp. is reported to be the functional bacteria in partial denitrification–anammox processes [25,26]; however, the OTUs for this species decrease to very low levels in the stored SWW. Consistently, the only dominant species in SWW_60_, i.e., *Halomonas*, and SWW_180_, i.e., *Vulcanibacillus,* are denitrifiers [27,28], while *Nitrosomonas*, dominant in SWW_120_, are the most well-known ammonia-oxidising bacteria [29].

Typical bacteria responsible for hydrolysis, acidification, and methane production, such as *Anaerovorax* species [30] and *Methanosaeta* [25], were detected as the dominant anaerobic bacteria in SWW. *Paenibacillus* became the most dominant species in SWW_180_, likely because it can survive for long periods under stress conditions and has a strong capacity for degrading different pollutants in wastewater systems [16]. Overall, the bacterial species observed as dominant in the SWW are denitrifiers, ammonia-oxidisers, and methane-producers.

Five suspected pathogenic bacterial species infecting animals and humans were detected in the SWW (Figure 6), and their identities are similar to that noted in a previous study [3]. Among the suspected pathogens, *Escherichia–Shigella* spp. play important role in the pathogenesis of diarrhoea [31]; *Vibrio* infections result in 8000 infections and 60 deaths each year in the USA (https://microbewiki.kenyon.edu/index.php/Vibrio accessed on 6 August 2010); *Arcobacter* spp. are classified by the International Commission on Microbial Specifications for Foods (ICMSF) as emerging pathogens [32]; *Clostridium* spp., a rapid-growing pathogen known to secrete an arsenal of 20 virulent toxins, has been associated with intestinal diseases in both animals and humans throughout the past century [33]; and some species of *Pseudomonas* spp. are among the most common opportunistic pathogens in nosocomial infections [34]. The relative abundance of both *Escherichia–Shigella* spp. and *Vibrio* spp. significantly increased with storage time (Figure 6a), while the relative abundances of *Arcobacter* spp., *Clostridium* spp., and *Pseudomonas* spp. fluctuated during the storage period (Figure 6b). The results indicate that these suspected bacterial pathogens persisted or were even enriched during the storage period.

As shown in Figure 7, *Ascomycota*, *Basidiomycota*, and *Neocallimastigomycota* were dominant fungi at phylum level in the SWW. The results are in agreement with those of Zhai et al. (2018) [5].

The top ten fungal genera were *Fusarium* spp. (2.70%), *Candelaria* spp. (1.47%), *Pseudobensingtonia* spp. (0.67%), *Neocallimastix* spp. (0.58%), *Ustilago* spp. (0.39%), *Geotrichum* spp. (0.27%), *Pleomonodictys* spp. (0.22%), *Blumeria* spp. (0.17%), *Mucor* spp. (0.17%), and *Thermomyces* spp. (0.15%). Interestingly, suspected fungal pathogens infecting plants were detected in SWW in this study. Fusarium is a genus of filamentous fungi that contains many agronomically important plant pathogens, mycotoxin producers, and opportunistic human pathogens [34]. The fungus Ustilago is a member of the smut fungi, a large group of parasites infecting mostly grasses, including several important crop plants such as maize, wheat, barley, and sugar cane [35]. Powdery mildew, caused by *Blumeria graminis f*. sp. tritici (Bgt), is one of the most destructive diseases that pose a great threat to wheat production [36]. Animal and human pathogens in SWW have been extensively investigated in previous studies [2,3,5]. In contrast, plant pathogens in SWW have been much less investigated. Some threatening fungal species for crops were detected in this study, indicating that SWW could pose an imminent threat to agriculture if reused as manure or irrigation without appropriate treatment. The results of this study indicate that fungi, including fungal pathogens, could be completely removed by anaerobic storage, thus substantially lowering the risks for SWW reuse.

## 4. Conclusions

Substantial losses of C and N were observed during SWW storage. The bacterial species found to be dominant in SWW after 180 days of anaerobic storage were denitrifiers, ammonia oxidisers, and methane producers. The bacterial and fungal abundances significantly decreased with storage time, which may be mainly attributed to nutrient loss and the long period of exposure to inhibitory antibiotics in the SWW. The suspected animal and human pathogens (e.g., *Escherichia–Shigella* spp., *Vibrio* spp., *Arcobacter* spp., *Clostridium_sensu_stricto_1* spp., and *Pseudomonas* spp.) and sulfonamide-resistant genes *Sul1*, *Sul2*, *Sul3*, and *SulA* tended to persist and even become enriched during SWW storage. *Fusarium* spp., *Ustilago* spp., and *Blumeria* spp., which are suspected fungal crop pathogens, were detected in SWW. The abundance of fungi, including suspected fungal pathogens, decreased to zero after 60 days of anaerobic storage. The results demonstrate that storage time is a crucial factor influencing the properties of SWW, and long periods of storage could lead to significant nutrient loss and enrichment of suspected bacterial pathogens and ARGs in SWW.

## Figures and Tables

**Figure 1 ijerph-20-03135-f001:**
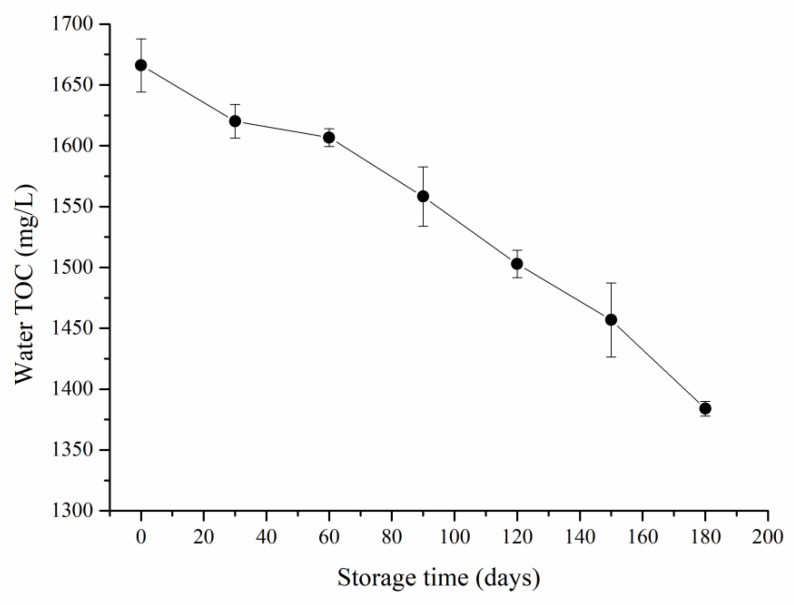
The swine wastewater total organic carbon (TOC) content at different storage times. Data are the means of three replications, and bars indicate standard error.

**Figure 2 ijerph-20-03135-f002:**
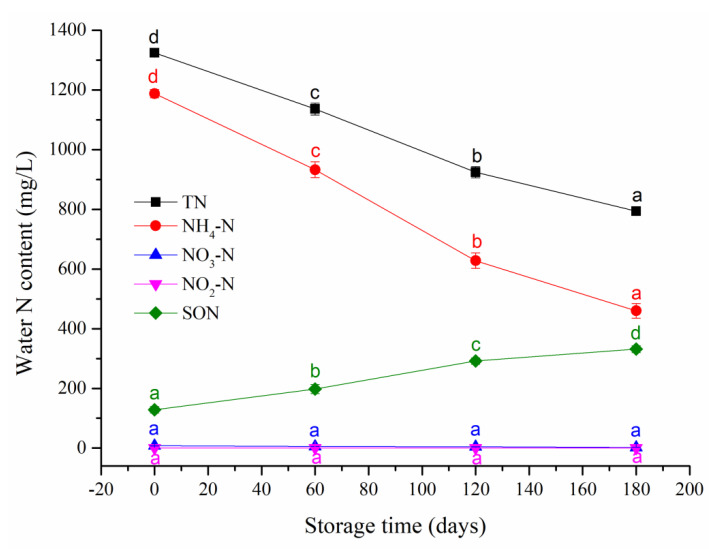
Contents of total nitrogen (TN), NH_4_-N, NO_3_-N, NO_2_-N, and soluble organic nitrogen (SON) in swine wastewater at different storage times. Data are the means of three replications, and bars indicate standard deviation. Different letters indicate significance at *p* < 0.05.

**Figure 3 ijerph-20-03135-f003:**
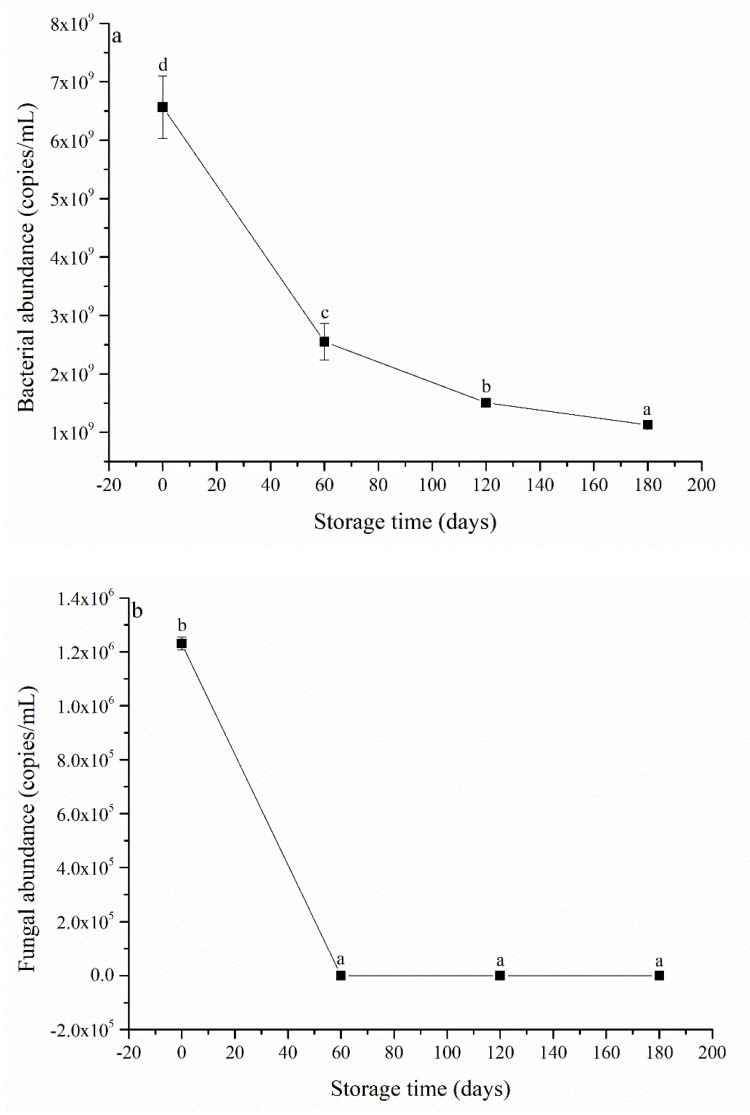
The bacterial (**a**); and fungal (**b**) abundances in swine wastewater at different storage times. Data shown are the means of three replicates, and different letters indicate significance at *p* < 0.05.

**Figure 4 ijerph-20-03135-f004:**
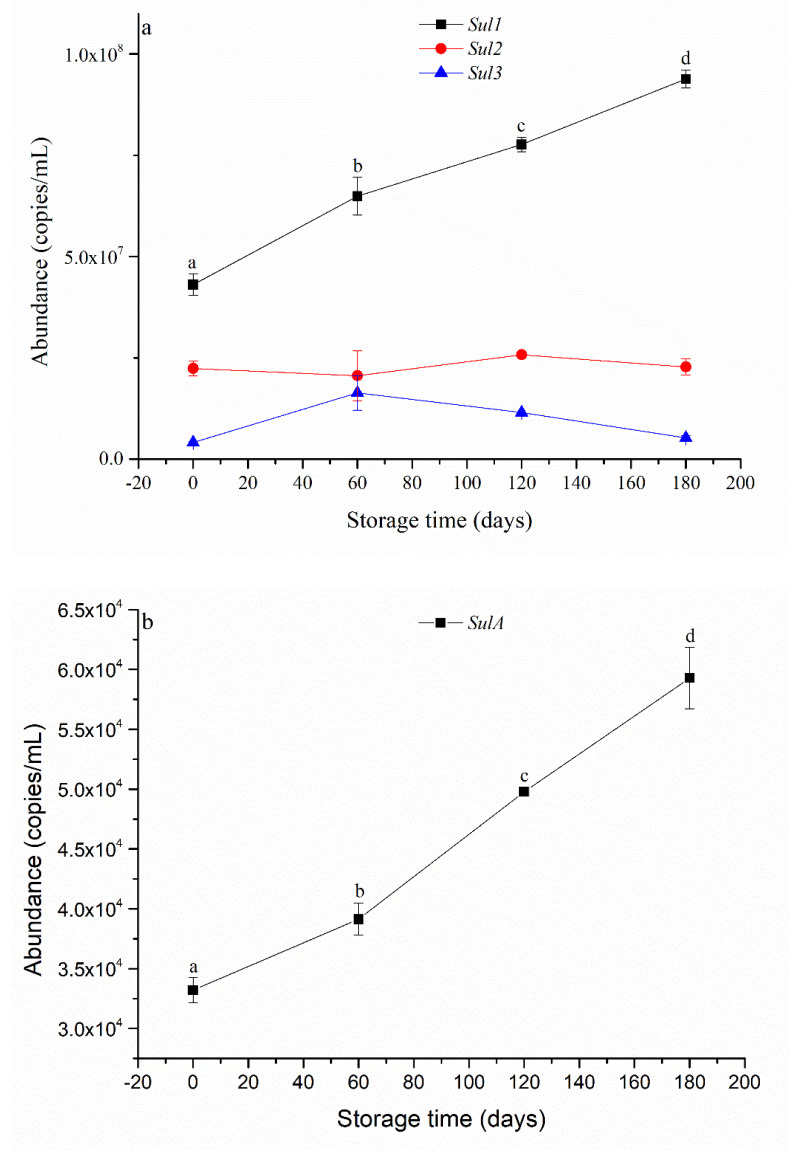
The abundances of *Sul1*, *Sul2*, and *Sul3* (**a**); and *SulA* (**b**) in swine wastewater at different storage times. Data shown are the means of three replicates, and different letters indicate significance at *p* < 0.05.

**Figure 5 ijerph-20-03135-f005:**
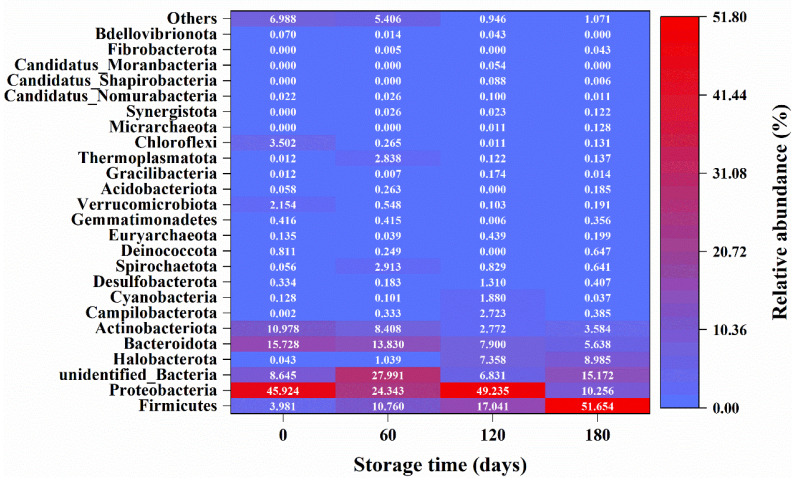
The relative bacterial abundance at phylum level on storage days 0, 60, 120, and 180 in swine wastewater. Data are the means of three replications.

**Figure 6 ijerph-20-03135-f006:**
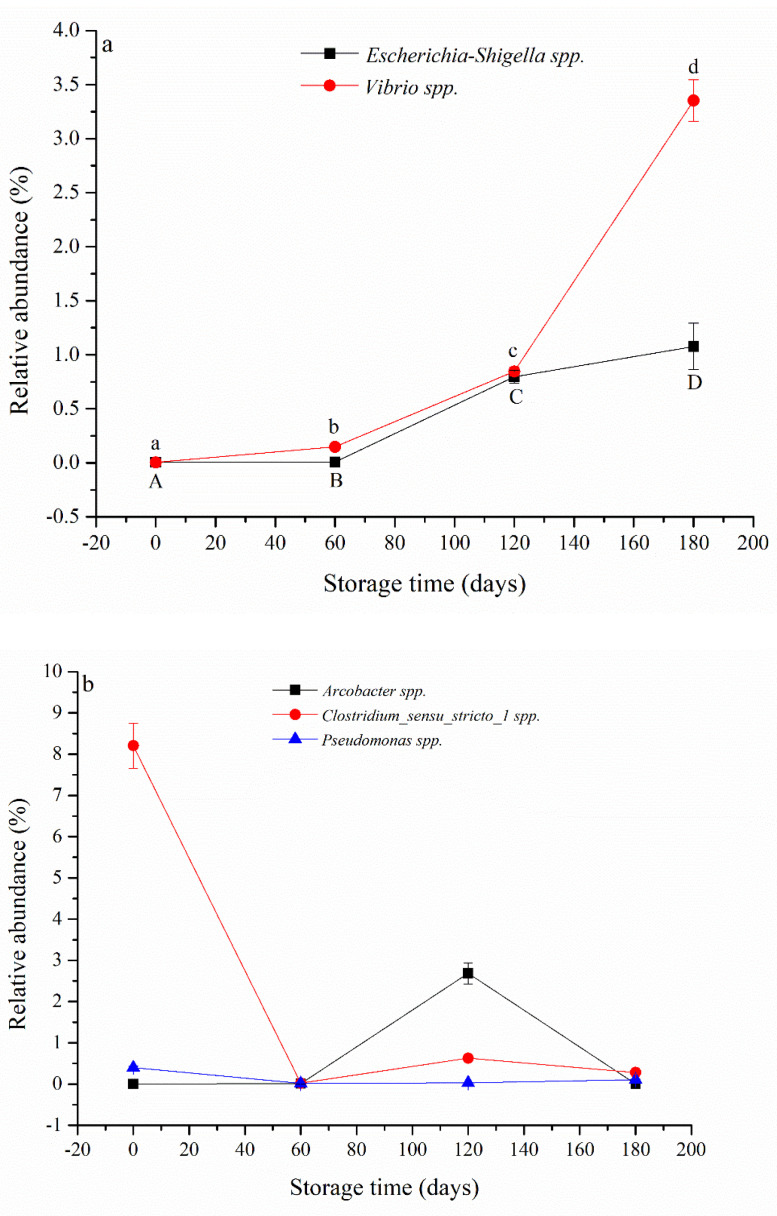
The relative abundance of suspected pathogenic bacterial species on storage days 0, 60, 120, and 180 in swine wastewater. Data are means of three replications, and different letters indicate significance at *p* < 0.05.

**Figure 7 ijerph-20-03135-f007:**
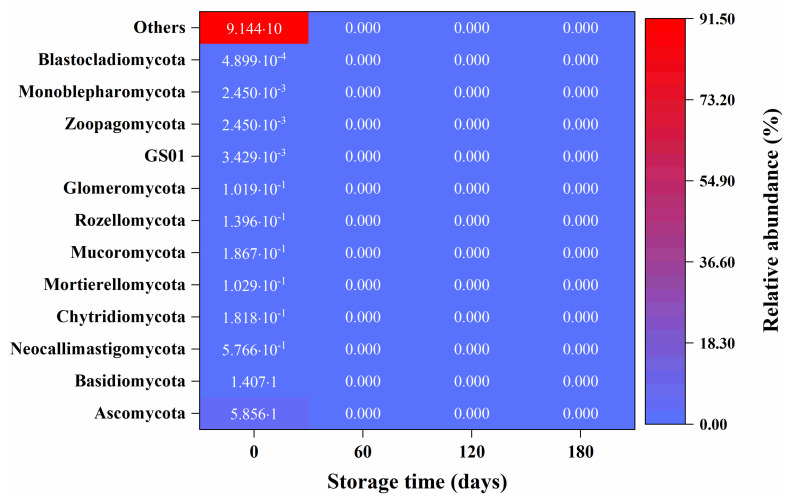
The relative fungal abundance at phylum level on storage days 0, 60, 120, and 180 in swine wastewater. Data are the means of three replications.

**Table 1 ijerph-20-03135-t001:** The properties of the raw swine wastewater in this study. Data are shown as means ± SD of three replications. TOC—total organic carbon; NH_4_-N—ammonia nitrogen; SAs—sulfonamides.

pH	TOC (mg/L)	NH_4_-N (mg/L)	SAs (μg/L)
8.23 ± 0.02	1666.1 ± 21.8	1187.5 ± 14.2	4653.2 ± 33.4

## Data Availability

The sequence data in this study have been saved in Sequence Read Archive (Accession number: PRJNA932233 and PRJNA932245). The public data can be accessed via https://www.ncbi.nlm.nih.gov/bioproject/PRJNA932233.

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
