# Peer review of "Anaerobic Storage Completely Removes Suspected Fungal Pathogens but Increases Antibiotic Resistance Gene Levels in Swine Wastewater High in Sulfonamides"

_ijerph, 2023, doi:10.3390/ijerph20043135_

Round 1
Reviewer 1 Report
Sample storage can be a cheap alternative to use in developing countries where wastewater is heavily relied upon for irrigation as a cheap source of nutrients. Therefore, this study is significant; however, significant revisions are required prior to publication.
English language correction is required. Please have a native speaker or reviewer review it. Simple errors such as two full stops (period sign) and no spacing after a full stop should be corrected.
Overall, introduction lacks significant background information. Information on mechanisms of how pathogens are eliminated by storage is not mentioned.
line 31-33. Bacterial and fungal pathogens are of great concern. But virus pathogens and helminths can also be present. Any reason to exclude viruses and helminths should be discussed.
Line 39. please correct. The presence of pathogens in swine wastewater does not contaminate food. It is the use of inadequately treated wastewater for irrigation that can contaminate food.
Line44-45 Why is storage of wastewater a compulsion? Please also specify the number of days. This can also help in discussion as the authors show 60 days are required to remove fungi. So the question of whether regulation time period is adequate or inadequate in removing pathogens is crucial to improve the significance of the study.
Line 47. The aim was to investigate the time required to remove pathogens by anaerobic storage. Please rephrase and adequately describe the objectives.
Line 52. What was the volume of wastewater collected? How much was filtered and what was the pore size of the filter paper? It is important because wastewater causes clogging and filtering large volumes (25L) is difficult. THe container was for 25L so was 25 L used?
line 59- The authors mentioned that samples were collected from the storage tank every 30 or 60 days. But the results show samples were collected every 60 days. Why was 60 days chosen? It is a long period of time. Short interval would have helped delineate the number of days required to remove pathogens.
Line 62-63. The authors need to describe the method clearly. They have mentioned filtration but mention supernatant. After filtration as the concentrating step, was bacteria and fungi eluted and supernatant analyzed?Its unclear and please mention clearly.
Line 66. Sulfonamides has been given a short form as Sul but the authors mentioned it later in Table1 as SA. It is not appropriate to use short form in for a single word unless its globally accepted. Please use the term "sulfonamides".
The whole Methodology section is highly questionable. Was DNA not extracted? The authors say 30 mL and 2 mL were used for sequencing and qPCR. Wastewater contain inhibitors. The presence of inhibitors is also not tested here. qPCR efficiency has not been mentioned. Please cite the necessary articles that developed primers. Quantification of pathogenic bacteria has been reported in results but it has not been described in methodology.
Line 77. ITS gene???
Line 99-100. The authors mention that decrease in bacterial abundances was due to high concentration of sulfonamides. Sulfonamides are antibiotics and can inhibit growth of bacteria or certain fungi maybe but not degrade DNA. PCR can detect 16S rRNA from dead or alive bacteria so this discussion is unacceptable.
Line 109. sula is mentioned but in Fig 2 sulA is mentioned. sulA in italics is correct and should be corrected.
Line 113. FCA has not been described.
Overall significant changes in abstract and discussion on valid points are required.
Author Response
Thanks for the valuable comments.The detailed responses are as follows:
Response: In the revised MS, we have stated the required storage time in different countries and hypothesized that storage time could crucially influence wastewater properties.
Line 47. The aim was to investigate the time required to remove pathogens by anaerobic storage. Please rephrase and adequately describe the objectives.
Response: The objectives in this study have been described as suggested in the revised MS.
Line 52. What was the volume of wastewater collected? How much was filtered and what was the pore size of the filter paper? It is important because wastewater causes clogging and filtering large volumes (25L) is difficult. THe container was for 25L so was 25 L used?
Response: A phrase“ through a 120-μm filtration paper” has been added in the revised MS.There was clear description of wastewater volume (“25-L working volume”, Line 62) in the MS.
line 59- The authors mentioned that samples were collected from the storage tank every 30 or 60 days. But the results show samples were collected every 60 days. Why was 60 days chosen? It is a long period of time. Short interval would have helped delineate the number of days required to remove pathogens.
Response: Some properties (e.g. N) was measured every 30 d, which has been presented in the revised MS.We chose 60 d as the measurement interval as 60 d was the least required storage time.
Line 62-63. The authors need to describe the method clearly. They have mentioned filtration but mention supernatant. After filtration as the concentrating step, was bacteria and fungi eluted and supernatant analyzed?Its unclear and please mention clearly.
Response: “supernatant” has been changed as “filtrate” in the revised MS.
Line 66. Sulfonamides has been given a short form as Sul but the authors mentioned it later in Table1 as SA. It is not appropriate to use short form in for a single word unless its globally accepted. Please use the term "sulfonamides".
Response: changed as suggested in the revised MS.
The whole Methodology section is highly questionable. Was DNA not extracted? The authors say 30 mL and 2 mL were used for sequencing and qPCR. Wastewater contain inhibitors. The presence of inhibitors is also not tested here. qPCR efficiency has not been mentioned. Please cite the necessary articles that developed primers. Quantification of pathogenic bacteria has been reported in results but it has not been described in methodology.
The sequence procedures were described in more detail in the revised MS.Quantification of pathogenic bacteria and fungi by relative abundance was described in section 2.6.
Line 77. ITS gene???
Response: It has been removed in the revised MS.
Line 99-100. The authors mention that decrease in bacterial abundances was due to high concentration of sulfonamides. Sulfonamides are antibiotics and can inhibit growth of bacteria or certain fungi maybe but not degrade DNA. PCR can detect 16S rRNA from dead or alive bacteria so this discussion is unacceptable.
Response: The RNA and DNA in dead bacteria and fungi could rapidly degrade, which was confirmed by that the fungi could not be detected after 60 d of storage.In the revised MS, we also discussed the effects of nutrient loss.
Line 109. sula is mentioned but in Fig 2 sulA is mentioned. sulA in italics is correct and should be corrected.
Response: corrected as suggested.
Line 113. FCA has not been described.
Response: ARB has been changed as antibiotic resistant bacteria.FCA is a class of antibiotics including several antibiotics.
Overall significant changes in abstract and discussion on valid points are required
Response: In Abstract, Discussion and Conclusion, statements about effects of storage time on pathogens and ARGs have been added in the revised MS.We also presented SWW nutrient data in the Results, and discussed the influencing mechanisms in-depth.
Reviewer 2 Report
The manuscript investigated the changes of pathogens and antibiotic resistance genes in anaerobic storage of antibiotics-high swine wastewater pathogens and antibiotic resistance genes. The research is meaningful, and it is suggested to hire after modification.
1. The expression of ammonia nitrogen should be NH4+-N instead of NH4-N, please modify.
2. The nicrobial abundances in line 96 are incorrectly stated, please correct.
3. Incorrect punctuation, e.g. lines 36, 49, 66 and 121. Please also check the full text.
4. Is the test method for bacteria abundances and fungus abundances the same? If there are different, please explain in Materials and methods.
5. Figure 5 is placed in the middle of a paragraph, please modify.
6. The language needs further polishing.
Author Response
Thanks for the valuable comments.The responses are as follows:
The manuscript investigated the changes of pathogens and antibiotic resistance genes in anaerobic storage of antibiotics-high swine wastewater pathogens and antibiotic resistance genes. The research is meaningful, and it is suggested to hire after modification.
- The expression of ammonia nitrogen should be NH4+-N instead of NH4-N, please modify.
Response: Either NH4+-N or NH4-N can be correct.
- The nicrobial abundances in line 96 are incorrectly stated, please correct.
Response: Corrected.
- Incorrect punctuation, e.g. lines 36, 49, 66 and 121. Please also check the full text.
Response: Corrected.The punctuation has been checked throughout the revised MS.
- Is the test method for bacteria abundances and fungus abundances the same? If there are different, please explain in Materials and methods.
Response: The methods have been described in more detail in the revised MS.
- Figure 5 is placed in the middle of a paragraph, please modify.
Response: modified as suggested.
- The language needs further polishing.
The English in the revised MS has been reviewed by my colleague.
Reviewer 3 Report
The manuscript " Anaerobic Storage Completely Removes the Fungal Pathogens but Increases the Antibiotic Resistance Gene Levels in Antibiotics-High Swine Wastewater" discusses the impact of storage time on fungal abundances and increase in ARG in swine waste. The authors have used microbial techniques to quantify ARGs and bacterial and fungal abundances. Improvements to the materials and methods sections should be made to give more details to the reader. The results and discussion section could also be improved with a more in-depth discussion on the possible reasons for the decrease in fungal abundance and increase in ARGs. The manuscript has several typographical errors with inconsistent spacing. In-text citations do not match the references. Specific comments are given below:
Fix typos in lines 36, 66, 121, and 206 and inconsistent spacing in several places throughout the manuscript.
In-text citation numbers do not match the references. For example, line 73 has Shi et al as [19], but the references section has a different paper.
The methods section should have more detailed information on the microbial analysis, including the kit used for DNA extraction, and the 16s region targeted for high throughput sequencing. Currently, there is no information on the type of high throughput sequencing technology used or the target region.
How was the high-throughput sequencing data processed?
In line 168, the authors discuss the relative fungal abundance of Escherichia-Shigella spp. and Vibrio spp, Arcobacter spp., etc. It is confusing how or why the authors discuss fungal abundances when the above-listed organisms belong to the bacterial domain.
In line 172, pathogen misspelled
Figure 5 represents the relative fungal abundance. Share more information on how these relative abundances were calculated. The relative abundances of different fungal phyla went down to 0. How was it concluded that the abundance was 0? Were they not detected in samples collected at the later storage time?
Author Response
Thanks for the valuable comments.The detailed responses are as follows:
The manuscript " Anaerobic Storage Completely Removes the Fungal Pathogens but Increases the Antibiotic Resistance Gene Levels in Antibiotics-High Swine Wastewater" discusses the impact of storage time on fungal abundances and increase in ARG in swine waste. The authors have used microbial techniques to quantify ARGs and bacterial and fungal abundances. Improvements to the materials and methods sections should be made to give more details to the reader. The results and discussion section could also be improved with a more in-depth discussion on the possible reasons for the decrease in fungal abundance and increase in ARGs. The manuscript has several typographical errors with inconsistent spacing. In-text citations do not match the references. Specific comments are given below:
Fix typos in lines 36, 66, 121, and 206 and inconsistent spacing in several places throughout the manuscript.
Response: Corrected as suggested.The punctuation has been checked throughout the revised MS.
In-text citation numbers do not match the references. For example, line 73 has Shi et al as [19], but the references section has a different paper.
Response: The match of in-text citation numbers with the references has been checked in the revised MS.
The methods section should have more detailed information on the microbial analysis, including the kit used for DNA extraction, and the 16s region targeted for high throughput sequencing. Currently, there is no information on the type of high throughput sequencing technology used or the target region.
How was the high-throughput sequencing data processed?
Response:The methods have been described in more detail in the revised MS.
In line 168, the authors discuss the relative fungal abundance of Escherichia-Shigella spp. and Vibrio spp, Arcobacter spp., etc. It is confusing how or why the authors discuss fungal abundances when the above-listed organisms belong to the bacterial domain.
Response: This paragraph discusses bacterial pathogens.
In line 172, pathogen misspelled
Response: corrected in the revised MS.
Figure 5 represents the relative fungal abundance. Share more information on how these relative abundances were calculated. The relative abundances of different fungal phyla went down to 0. How was it concluded that the abundance was 0? Were they not detected in samples collected at the later storage time?
Response: The relative abundance for each taxon was calculated as the operational taxonomic unit (OTU) ratio (%) among the total OTUs, which was stated in section 2.6.Fungi were not detected at later stages.
Round 2
Reviewer 1 Report
English corrections are required. I strongly recommend english language corrections with a help of native speaker. Minor corrections such as space after full stops are to be checked. Specific comments are mentioned below.
Title- The authors mention in title that wastewater studied were high in antibiotics but in Line 111 wrote that only sulfonamides were used in the pig farms . High antibiotics could mean high concentration of any antibiotics and only sulfonamides were confirmed by measurements. Please be specific.
Line 37. Please check the word "particularly".
Line 44. Please rephrase. Swine wastewater does not reinfect but the pathogens present will reinfect.
Line 50. The required time varies in these countries. It is very important as background information as to why these days were implemented. Please mention why it varies between countries because the aim of this study is to delineate time frame for adequate removal of pathogens.
Line 91-92. How was bacterial DNA extracted from 30 mL of wastewater. There was no concentrating step required. What was done to 30 mL prior to extraction? Please be kind enough to mention it.
Line 102. Was 2 mL of wastewater was directly subjected to qPCR? Wastewater are often prone to presence of inhibitors. Were the presence of inhibitors assessed before qPCR?
Line 104. please check "16S rRNA".
Line 126-129. Please rephrase and write in scientific ways to report results. Please mention p values and tests used in brackets whenever significant is mentioned throughout the manuscript. If test used is same, at least be kind enough to mention p-values.
Line 149 and 152. The term antibiotics encompasses many different classes of antibiotics. Since sulfonamides were only used in the farms, these sentences needs to be rephrased.
Line 164. I had previously requested to describe FCA. There is no universally accepted antibiotic class as FCA. Please read Pu et al. which you have cited and describe what FCA is as the author might have made a short form of three classes of antibiotics.
Line 248. Please do not write "some fungal species". It is your result and can be of high significance. Please mention which fungi that infects plants were found clearly. If all has already been mentioned ahead in earlier lines then rephrase. If more are to be added please include specific names.
Line 259 The authors used 180 days here but used "d" throughout the manuscript. Please minimize the use of initials as it can confuse readers. Whenever it is one word, please use the full form as it makes no difference to words count.
Author Response
Many thanks for the valuable comments.We provide detailed responses as follows:
English corrections are required. I strongly recommend english language corrections with a help of native speaker. Minor corrections such as space after full stops are to be checked. Specific comments are mentioned below.
Response: The English been edited by MDPI Language Editing Services in the revised MS.
Title- The authors mention in title that wastewater studied were high in antibiotics but in Line 111 wrote that only sulfonamides were used in the pig farms . High antibiotics could mean high concentration of any antibiotics and only sulfonamides were confirmed by measurements. Please be specific.
Anaerobic Storage Completely Removes Fungal Pathogens but Increases Antibiotic Resistance Gene Levels in Swine Wastewater High in Sulfonamides”.
Response: The title has been changed as “
Line 37. Please check the word "particularly".
Response: Checked.
Line 44. Please rephrase. Swine wastewater does not reinfect but the pathogens present will reinfect.
Response: Changed as suggested in the revised MS.
Line 50. The required time varies in these countries. It is very important as background information as to why these days were implemented. Please mention why it varies between countries because the aim of this study is to delineate time frame for adequate removal of pathogens.
Response: A statement “ This variation in the required storage time may result from differences in the standard and wastewater properties in these countries” has been added in the revised MS.
Line 91-92. How was bacterial DNA extracted from 30 mL of wastewater. There was no concentrating step required. What was done to 30 mL prior to extraction? Please be kind enough to mention it.
Response:A description “Before the DNA extraction, the sample was centrifugated at 300 g for 5min to precipitate large particles. The supernatant was then removed and centrifuged at 19000 g for 5min to precipitate cells. Moreover, the pellet was double washed with 3.0 ml of sterile distilled water to reduce inhibitors for Taq polymerase” has been added in the revised MS.
Line 102. Was 2 mL of wastewater was directly subjected to qPCR? Wastewater are often prone to presence of inhibitors. Were the presence of inhibitors assessed before qPCR?
Response: The procedure has been rewritten in the revised MS.
Line 104. please check "16S rRNA".
Response: changed as suggested in the revised MS
Line 126-129. Please rephrase and write in scientific ways to report results. Please mention p values and tests used in brackets whenever significant is mentioned throughout the manuscript. If test used is same, at least be kind enough to mention p-values.
Response: “(P < 0.05)” has been added after “significant” or “significantly” in these senstences.
Line 149 and 152. The term antibiotics encompasses many different classes of antibiotics. Since sulfonamides were only used in the farms, these sentences needs to be rephrased.
Response: “antibiotics” has been replaced with “sulfonamides” in the revised MS.
Line 164. I had previously requested to describe FCA. There is no universally accepted antibiotic class as FCA. Please read Pu et al. which you have cited and describe what FCA is as the author might have made a short form of three classes of antibiotics.
Response: “cefaclor” has been added before “FAC” in the revised MS.
Line 248. Please do not write "some fungal species". It is your result and can be of high significance. Please mention which fungi that infects plants were found clearly. If all has already been mentioned ahead in earlier lines then rephrase. If more are to be added please include specific names.
Response: The sentences are rephrased as “Fusarium spp., Ustilago spp., and Blumeria spp., which are fungal crop pathogens, were detected in SWW” in the revised MS.
Line 259 The authors used 180 days here but used "d" throughout the manuscript. Please minimize the use of initials as it can confuse readers. Whenever it is one word, please use the full form as it makes no difference to words count.
Response: “d” is replaces as days throughout the MS in the revised MS.
Reviewer 3 Report
The authors have satisfactorily addressed all comments. It can be accepted for publishing.
Author Response
Response: Thanks.